# Dietary Pyrethroid Exposures and Intake Doses for 188 Duplicate-Single Solid Food Items Consumed by North Carolina Adults

**DOI:** 10.3390/toxics8010006

**Published:** 2020-01-22

**Authors:** Marsha K. Morgan

**Affiliations:** United States Environmental Protection Agency’s Center for Public Health and Environmental Assessment, Research Triangle Park, NC 27711, USA; morgan.marsha@epa.gov; Tel.: +1-919-541-2598

**Keywords:** insecticides, food, diet, humans, exposure, intake dose, risk assessment

## Abstract

Few studies have measured pyrethroid residue concentrations in food items consumed by adults in their daily environments. In a further analysis of study data, the objectives were to determine pyrethroid residue levels in single, solid food items consumed by adults and to estimate dietary pyrethroid exposures and intake doses per food item. A total of 50 adults collected 782 duplicate-diet solid food samples over a six-week monitoring period in North Carolina between 2009 and 2011. Of these samples, 188 contained a single, solid food item (i.e., lasagna). Levels of eight pyrethroids were quantified in the 188 food items using LC–MS/MS. At least one pyrethroid was detected in 39% of these food items. *Cis*-permethrin (17%), bifenthrin (15%), *trans*-permethrin (14%), and deltamethrin (14%) were detected the most often. Cyfluthrin, cyhalothrin, cypermethrin, and esfenvalerate were all detected in <6% of the samples. The highest residue level was found in a pizza sample containing both *cis*-permethrin (96.4 ng/g) and *trans*-permethrin (73.7 ng/g). For *cis*-permethrin, median residue levels (≥LOQ) were significantly higher (*p* = 0.001) in foods that contained a fruit/vegetable compared to foods that did not. For individual pyrethroids, the participants’ maximum dietary intake doses in the single food items ranged from 38.1 (deltamethrin) to 939 ng/kg/day (*cis*/*trans*-permethrin).

## 1. Introduction

In the United States (US), the synthetic pyrethroid insecticides have been widely used to control insects in agricultural settings since the 1970s [1,2]. More than 20 different pyrethroids are currently approved by the US Environmental Protection Agency (EPA) for agricultural use [3]. It is estimated that on average ~40 million acres of farmland each year are treated with pyrethroids to kill insect pests primarily on 11 major food crops including citrus, corn, potatoes, rice, sorghum, soybeans, sugar beets, sunflowers, sweet corn, tomatoes, and wheat [4]. Pyrethroids (i.e., cyfluthrin and deltamethrin) are also commonly applied in empty grain farm bins or on grains prior to storage in farm bins to prevent insect infestations [5,6,7]. In addition, livestock are routinely treated with products containing pyrethroids (e.g., sprays, pour-ons, dips, or ear tags) to control insects on farms [1,8].

Research has suggested that the diet is a major source of exposure to pyrethroids in US adults [3,9,10,11]. In the US, the Food and Drug Administration’s (FDA’s) Total Diet Study (TDS) is used to identify food items that contain detectable pesticide residues, including the pyrethroids, which are commonly consumed by the general population [9]. Food items in the TDS are purchased from grocery stores and supermarkets across the country and then prepared ‘as consumed’ and analyzed for pesticides in a laboratory [9]. In the latest 2012 TDS, pyrethroid residues were found in a variety of food items (e.g., vegetables, fruits, cereals, breads, pizzas, spaghetti, lasagna, salmon, and tuna noodle casserole) [9]. Few US studies, however, have measured the concentrations of pyrethroid residues in similar food items that are prepared and/or consumed by adults in their daily environments [10,12,13]. This type of study design is important as people’s personal behaviors and hygiene (e.g., not cleaning kitchen countertops before preparing foods on them or not rinsing fresh produce with tap water before consumption) can greatly impact pesticide residue levels in some food items at home or other settings (i.e., restaurants) [12]. At the moment, it is unclear whether pesticide residue levels differ substantially in commonly consumed food items of adults measured in the FDA’s TDS compared to ‘real world’ studies.

After consumption of food, the lipophilic pyrethroids are rapidly absorbed and metabolized and are mainly renally excreted with urinary elimination half-lives generally <15 h [14]. Several recent cross-sectional studies have raised concerns that environmental exposures to pyrethroids may be causing adverse health effects (i.e., neurological, endocrine, reproductive, and cardiac) in adults [15,16,17], but temporal studies are needed to confirm these results [18].

In prior work from the “Pilot Study to Estimate Human Exposures to Pyrethroids using an Exposure Reconstruction Approach” (Ex-R study), the concentrations of eight current-use pyrethroids (bifenthrin, cyfluthrin, cyhalothrin, cypermethrin, deltamethrin, esfenvalerate, *cis*-permethrin, and *trans*-permethrin) were quantified in 782 duplicate-diet solid food (DDSF) samples of 50 North Carolina adults that were collected on days 1 and 2 during weeks 1, 2, and 6 over a six-week monitoring period [13]. For each participant, the DDSF samples (up to 18) were collected and homogenized over three consecutive time periods (04:00–11:00, 11:00–17:00, and 17:00–04:00) each sampling day. The results showed that 49% of the 782 DDSF samples contained one or more of the target pyrethroids [13]. However, at that time, it was unknown which consumed single food items (e.g., bagel, pizza) likely contributed the most to the adults’ estimated dietary exposures and intake doses to these insecticides. In a further analysis of the study data, using the Ex-R participants’ food diaries, the records showed that 188 out of the 782 DDSF samples contained only a single, solid food item. The objectives for this current work were (1) to determine the residue levels of the individual target pyrethroids in each single solid food item and (2) to estimate the participant’s dietary exposures and intake doses for each pyrethroid per food item.

## 2. Materials and Methods

### 2.1. Study Background

The study design, sampling methodology, and human subject protection approval by the University of North Carolina’s Institutional Review Board in 2008 (study number 09-0741) were described in-depth in Morgan et al. [19]. Briefly, in 2009–2011, the Ex-R study was performed at the US EPA’s Human Studies Facility (HSF) in Chapel Hill, North Carolina, and at the participants’ residences (≤40-mile radius of this facility). Adult participants were recruited by an EPA contractor using an existing database of potential volunteers or by word of mouth (e.g., previous study participants) [13]. A total of 30 females (ages 21–50) and 20 males (ages 19–48) were recruited [13]. The participants filled out 24 h food diaries and collected 24 h DDSF samples (excluding beverages) on days 1 and 2 during sampling weeks 1, 2, and 6. None of the participants reported working in occupations (e.g., factories, farms, and pest control services) that would have directly exposed them to pyrethroid residues at least one month prior and during the six-week monitoring period. All adults signed informed consent documents prior to participating in this study. The data presented in this article apply only to this study cohort and cannot be generalized to all adults in North Carolina or all adults in the US.

### 2.2. Collection of Food Diaries and Solid Food Items

The 24 h food diaries were completed over three consecutive sampling periods (period 1 = 04:00–11:00, period 2 = 11:00–17:00, and period 3 = 17:00–04:00) each sampling day [19]. For each sampling period, paper food diaries were used to record each consumed solid food item, including estimated amount (in cups), and to list any fruits or vegetables that were part of this item. In addition, the food diaries were used to record that each consumed solid food item was also part of the corresponding 24 h DDSF sample (see below).

The 24 h DDFS samples were also collected over the same three, consecutive sampling periods each sampling day (as mentioned above). DDSF samples were defined as duplicate amounts of each solid food item (prepared as consumed) that a person ate each sampling day [13]. Examples of solid food items were bagels, cookies, cereals with milk, fruit smoothies, ice creams, pizzas, salads, soups, spaghetti, and yogurts with cereal. The participants put duplicate amounts of all the solid food items that they ate during each sampling period into a prelabeled and re-sealable, polyethylene sampling bag (31 × 31 cm, Uline Shipping Supply Specialist^®^). The sampling bags were placed into the provided portable thermoelectric coolers (Vinotemp^®^ or Princess International^®^) to keep the food samples at reduced temperatures [19]. Temperature data loggers (Easy Log EL-USB-LITE or EL-USB-1, Lascar Electronic, Ltd.) were used to record the internal temperatures of the coolers during each sampling day.

The participants returned the portable coolers containing the food diaries and the DDSF samples to the HSF on day 3 of each sampling week [19]. At the HSF, a technician verified with each participant that the food diaries were properly filled out and duplicate amounts of each consumed solid food item were collected during the sampling period. In addition, the technician inspected each food sampling bag (i.e., for leaks, properly closed) and recorded the amount (g) of each bag using a calibrated, weight scale. Then, the technician transported the portable coolers containing the food diaries and food sampling bags (with blue ice) in a van to a US EPA laboratory in Research Triangle Park, North Carolina. The food sampling bags were stored at ≤20 °C in laboratory freezers until chemical analysis.

### 2.3. Chemical Analysis

The food sampling bags (*n* = 188) containing single solid food items were removed from the laboratory freezers and thawed to room temperature. The solid food items were separately homogenized using a vertical cutter mixer (R10-Ultra^®^ or Robot Coupe R4N-D^®^) or a high-speed blender (Waring MBB518^®^) for watery food items such as fruit smoothies, protein shakes, or soups [13]. The target pyrethroids were extracted from each homogenized food item using a QuEChERS method (quick, easy, cheap, effective, rugged, and safe) [13,20,21] modified for use in a variety of complex foods matrices [13]. Briefly, each food item (2 g) was placed into a glass vial, spiked with internal standards (^13^C_6_-λ-cyhalothrin, ^13^C_6_-cypermethrin, ^13^C_6_-*cis*-permethrin, and ^13^C_6_-*trans*-permethrin; Cambridge Isotope Laboratories, Andover, MA, USA), and 2 mL of acetonitrile was added. The sample was mechanically shaken (~2 min) and 200 mg of sodium chloride and 800 mg of magnesium sulfate was added. Then, it was vortexed until thoroughly mixed and centrifuged at 4000 rpm (5 min). Next, a 1 mL amount of the supernatant was filtered through an ENVI-carb (120/400) clean-up tube (Supelco, Sigma-Aldrich, Bellefonte, PA, USA). The extract was then vortexed (0.5 min) and centrifuged at 15,000 rpm (10 min). Finally, a 300 µL aliquot of the extract was diluted with 0.1% formic acid (1:1 ratio), filtered (0.2 µm PTFE filter), and refrigerated (~4 °C) until analysis.

The residue levels of the target analytes were concurrently quantified in each sample extract using electrospray ionization liquid chromatography–tandem mass spectrometry (LC–MS/MS) with a 4000 QTrap linear ion mass spectrometer (AB Sciex, Framingham, MA, USA) [13]. The limits of quantitation (LOQs) were 0.10 ng/g (bifenthrin), 2.5 ng/g (cyfluthrin), 1.0 ng/g (cyhalothrin), 1.0 ng/g (cypermethrin), 0.25 ng/g (deltamethrin), 1.0 ng/g (esfenvalerate), 0.25 ng/g (*cis*-permethrin), and 0.25 ng/g (*trans*-permethrin), respectively. The LOQs were based on the levels of the lowest calibration standards with a signal-to-noise ratio of not less than 3 [13]. For these analytes, the estimated limits of detection were 2 to 5 times lower than the estimated LOQs [13]. The quality assurance and quality control (QC) procedures and QC results for the collection and analysis of the food items was described in-depth in Morgan et al. [13], and the QC samples (i.e., blanks, spikes, and duplicates) met all data quality criteria.

### 2.4. Statistical Analysis

Using JMP software (2015 version 12.0.1, SAS Institute Inc., Cary, NC, USA) summary statistics (i.e., detection frequency, median, and range) were computed for each pyrethroid (≥LOQ) in the duplicate-single solid food items (*n* = 188). For individual pyrethroids detected ≥14% in the samples, an unpaired Mann–Whitney test (GraphPad Prism software, 2010 version 5.04, GraphPad Inc., La Jolla, CA, USA) was performed to determine whether there was a statistically significant difference (*p* < 0.05) in median residue levels between food items with at least one fruit or vegetable and food items without any fruits or vegetables. Only values at or above the LOQ for each pyrethroid was used in this analysis. Also prior to statistical analysis, the shapes of the distributions were determined to be similar between food groups for each pyrethroid (JMP software).

### 2.5. Estimation of Dietary Exposures and Intake Doses per Solid Food Item

The participant’s estimated dietary exposure (ng/day) for each pyrethroid in a single, solid food item was calculated by multiplying the concentration in the food item (ng/g) by the total mass of food (g) in the sampling bag. Then, the participant’s estimated dietary intake dose (ng/kg/day) was computed by dividing the dietary exposure estimate (ng/day) by the person’s body weight (kg) and assuming a default absorption rate of 100% in the gut. Food mass records were not available for 12 food items. For these samples, the food mass was estimated by using the reported amount consumed (i.e., 2 cups) in the participants’ food diaries. Body weights of the females ranged from 48.1 to 130 kg, and body weights of the males ranged from 57.6 to 130 kg [13].

## 3. Results

### 3.1. Pyrethroid Residue Levels in the Single Food Items as Consumed

Table 1 presents the concentrations of the individual target pyrethroids that were found in each duplicate-single solid food item. The results showed that at least one pyrethroid was detected in 73 out of the 188 food items (39%) consumed by Ex-R adults over the six-week monitoring period. The pyrethroids detected most often in the food items were *cis*-permethrin (17%), bifenthrin (15%), *trans*-permethrin (14%), and deltamethrin (14%). Using all 188 samples, residue levels for these four pyrethroids at the 90th percentile were 0.72 (*cis*-permethrin), 0.64 ng/g (*trans*-permethrin), 0.61 ng/g (bifenthrin), and 1.1 ng/g (deltamethrin). For these same pyrethroids (using values ≥ LOQ), median levels were 0.88 ng/g (*cis*-permethrin), 0.98 ng/g (*trans*-permethrin), 0.72 ng/g (bifenthrin), and 2.25 ng/g (deltamethrin). The highest residue levels for *cis*-permethrin (96.4 ng/g) or *trans*-permethrin (73.7 ng/g) occurred in the same sample of Greek pizza. For bifenthrin and deltamethrin, the greatest residue concentrations were found in a fruit smoothie with coconut milk (13.9 ng/g) and ginger snap cookies (16.3 ng/g), respectively. The other pyrethroids (cyfluthrin, cyhalothrin, cypermethrin, and esfenvalerate) were all detected in <6% of the food items. However, cypermethrin was among all the target pyrethroids that had the highest measured residues in the food items—which included an egg and sausage biscuit (64.2 ng/g), ginger snap cookies (63.3 ng/g), and a chicken biscuit (50.1 ng/g).

An unpaired Mann–Whitney test was run to compare residue levels (≥LOQ) for bifenthrin, deltamethrin, *cis*-permethrin, or *trans*-permethrin between food items with at least one fruit or vegetable and food items without any fruits or vegetables (Table 2). The results showed that there were no significant differences (*p* < 0.05) occurring for the levels of bifenthrin or *trans*-permethrin residues between the two types of food groups. However, median concentrations of *cis*-permethrin were significantly higher (*p* = 0.001) in foods that contained a fruit/vegetable (1.5 ng/g) compared to foods that did not contain a fruit/vegetable (0.38 ng/g). In addition, median deltamethrin residues were marginally significantly greater (*p* = 0.058) in food items without a fruit or vegetable (4.9 ng/g) compared to those with a fruit or vegetable (1.7 ng/g). 

### 3.2. Pyrethroid Co-Occurrence in the Single Food Items as Consumed

Figure 1 presents the pyrethroid insecticides that co-occurred in 33 out of the 77 different food items with detectable residues (≥LOQ). The results showed that 39% and 27% of these samples had total pyrethroid residue levels at or above 10.0 ng/g and 20.0 ng/g, respectively. Also, 39% of these food items contained a minimum of three different pyrethroids; *cis*-permethrin, *trans*-permethrin, and bifenthrin co-occurred the most often in these samples. In addition, 21% of these samples contained at least four different pyrethroids, namely bifenthrin, deltamethrin, *cis*-permethrin, and *trans*-permethrin. Only one sample, chicken pizza, contained five different pyrethroids (bifenthrin, cyfluthrin, cyhalothrin, *cis*-permethrin, and *trans*-permethrin); combined residue levels were 22.1 ng/g. Based on co-occurrence, the highest combined pyrethroid residue level was found in a Greek pizza sample containing *cis*-permethrin and *trans*-permethrin (170 ng/g) (see footnote a in Figure 1), followed by a ginger snap cookie sample containing cypermethrin and deltamethrin (79.6 ng/g).

### 3.3. Participant Pyrethroid Exposures and Intake Doses for Consumed Food Items

The adults’ estimated dietary exposures (ng/day) and intake doses (ng/kg/day) for the target pyrethroids detected in each food item are provided in Table 3. For the four most frequently detected pyrethroids (using samples ≥LOQ), the adults’ estimated median dietary exposures were 212 ng/day (*cis*-permethrin), 204 ng/day (*trans*-permethrin), 134 ng/day (bifenthrin), and 184 ng/day (deltamethrin). Among these pyrethroids, the highest dietary exposure occurred for *cis*-permethrin (47,040 ng/day), followed by *trans*-permethrin (35,980 ng/day) in the same sample of Greek pizza. For the other pyrethroids (Table 3), the highest dietary exposures were for cypermethrin in a fruit smoothie with milk (9660 ng/day) and cyhalothrin in tofu with vegetables (9560 ng/day).

For the most frequently detected pyrethroids (using samples ≥LOQ), the adults’ estimated median dietary intake doses were 2.7 ng/kg/day (*cis*-permethrin), 2.1 ng/kg/day (*trans*-permethrin), 1.6 ng/kg/day (bifenthrin), and 2.8 ng/kg/day (deltamethrin) (Table 3). The highest dietary intake oses were for *cis*-permethrin (532 ng/kg/day) and *trans*-permethrin (407 ng/kg/day) in the same Greek pizza sample. For the remaining pyrethroids, the greatest dietary intake doses occurred for cypermethrin in a fruit smoothie with coconut milk (98.2 ng/kg/day) and cyhalothrin in tofu with vegetables (97.1 ng/kg/day); these two samples were consumed by the same participant, but on different sampling days.

## 4. Discussion

Only a few studies, conducted in the US between 2005 and 2011 have measured pyrethroid residues in the actual foods consumed by adults in their daily environments [10,12,13]. In these prior studies, the participants’ food items were composited by either food group, meal type (breakfast, lunch, and dinner), or several time periods over a day. Because the samples were composited, a major weakness of these earlier studies was not being able to identify the specific food items (e.g., pizza) or components of these food items (e.g., spinach, tomatoes) that had measurable pyrethroid residues. In this current work, the levels of eight current-use pyrethroids were quantified in 188 duplicate-single solid food items consumed by Ex-R adults over a six-week monitoring period in North Carolina in 2009–2011. *Cis-*permethrin and *trans*-permethrin were detected in 17% and 14% of the food samples, respectively. The top food items that contained measurable total permethrin residues were various kinds of bagels, cereals (with and without milk), pizzas, and sandwiches (with cheese, eggs, bologna and/or ham). The highest residue levels for the two permethrin isomers were in a Greek pizza sample with spinach, tomatoes, olives, and onions and a lasagna sample with spinach and tomatoes (Table 1). An important study result was that the median concentrations of *cis*-permethrin (≥LOQ) were at least three times greater (*p* = 0.001) in food items that had at least one fruit and/or vegetable compared to those that did not. Also, 93% of the top 15 food samples with measurable *cis*-permethrin or *trans*-permethrin residues (above 1 ng/g) contained at least one fruit or vegetable (Table 3). Tomatoes were the most frequently found fruit in these samples (9 out of 15 food items), and carrots, lettuce, onions, raisins, peppers, and spinach were the most frequently found vegetables (3 out of 15 food items). In perhaps the most comparable study, the US FDA’s Total Diet Study (2009–2010) reported that these two permethrin isomers were the most frequently detected pyrethroids in a variety of food items purchased from grocery stores and supermarkets across the country [9]. In that study, the food items that contained the highest residue levels of these two permethrin isomers were raw or frozen spinach, lettuce, collards, and Brussel sprouts. In addition, in the 2009–2011 US Department of Agriculture’s (USDA’s) Pesticide Data Program (PDP) surveys, *cis*- and *trans*-permethrin residues were detected mainly in certain vegetables (i.e., spinach, lettuce, sweet bell peppers) [22]. This above information suggests that the consumption of certain types of produce either in raw or processed foods are likely a major dietary source of (total) permethrin exposure in US adults.

Few published data are available on the levels of bifenthrin residues in foods consumed by adults in their everyday environments [12,13]. Melynk et al. [12] reported finding bifenthrin residues in 30% of the 67 composited solid food samples (by meal type) of nine Hispanic women in Florida in 2008 (maximum residue = 135 ng/g). In earlier work by Morgan et al. [13], bifenthrin residues were detected in 20% of the 782 DDSF samples of 50 Ex-R adults (maximum residue = 13.9 ng/g). In this current work, bifenthrin residues were detected in 15% of the 188 duplicate-single solid food items consumed by Ex-R adults, and the maximum residue level of 13.9 ng/g was in a fruit smoothie with milk. The top food item that most often contained bifenthrin residues was various kinds of pizzas. Also, over 75% of all the food items with detectable bifenthrin residues contained at least one fruit or vegetable, and tomatoes (13 out of 22 samples) were found the most often in these samples. In comparison to the 2011 USDA’s PDP survey, bifenthrin residues were also detected the most frequently (19%) in tomato samples [22]. As data are limited, more research is needed to determine bifenthrin residue levels in raw and processed foods commonly consumed by adults.

In this study, deltamethrin residues (≥4.9 ng/g) were found the highest in processed food items that contained only dairy, grain and/or meat products. These items included ginger snap cookies (16.3 ng/g), plain biscuit (12.0 ng/g), cake (8.9 ng/g), chicken leg (6.6 ng/g), egg and sausage biscuit (6.1 ng/g), cheese and bologna sandwich (5.6 ng/g), and a chicken biscuit (4.9 ng/g). These data agree with Reiderer et al. [10] who also reported finding deltamethrin residues in composited food samples that only contained grains, dairy, or meat/fish/eggs items from 12 adults in Georgia USA in 2005–2006. In that study, the maximum deltamethrin residue of 389 ng/g occurred in a sample containing only a bagel, bread, muffin, and cake [10]. Research has suggested that a major source of deltamethrin residues in these food items is likely from post-harvest pesticide applications to grains (e.g., corn, oats, and wheat) in storage bins or pesticide applications to empty grain storage bins [5,6,7].

Few data are available in the literature on the co-occurrence of pyrethroids residues in composited food samples [10,12,13], and none in single, solid food items consumed by adults in their daily environments. In this current study, 39% of the 73 duplicate-single food items with detectable residues contained at least three different pyrethroid insecticides (Figure 1). The pyrethroids that co-occurred the most often in these samples were bifenthrin, *cis*-permethrin, and *trans*-permethrin, and total residue levels for these pyrethroids ranged from 1.3 to 40.4 ng/g in these samples. The top food items that most often had measurable residues of these three pyrethroids were various types of pizzas and bagels (range = 4.3–22.1 ng/g). More research is needed on the co-occurrence of pyrethroid residues in single food items that are commonly eaten by adults.

In the US, the FDA’s TDS measures pesticide residues in food items commonly eaten in the average diet of the general population [9]. These data are used to estimate the annual dietary intake of adults to specific pesticides, including pyrethroid insecticides [9]. The TDS purchases about 280 different kinds of food items from grocery stores and supermarkets from four geographical regions of the country each year; the foods are then prepared as consumed and analyzed for pesticides in a laboratory [9]. Few US studies, however, have measured pesticide residues in similar food items consumed by adults in their everyday environments [10,12,13]. This is important as research has indicated that people’s personal behaviors and hygiene (e.g., not washing hands before eating, not cleaning kitchen countertops before preparing foods on them, or not rinsing fresh produce with tap water before consumption) can greatly influence pesticide residue levels found in some foods [12,23,24,25]. For instance, studies have shown that pyrethroid residues have been found on kitchen countertops at homes [24,25], and these residues can be transferable to foods contacting these surfaces [26]. The major benefit of these ‘real world’ studies is that they account for possible increases or decreases in pesticide residue concentrations occurring in food items that are prepared and/or eaten by people at home or other places (i.e., work or school). In this current study, food items with the greatest deltamethrin residues (>6 ng/g) were gingersnap cookies, cake, a plain biscuit, chicken leg, and an egg and sausage biscuit. However, in comparison to the 2009–2012 TDS surveys, the highest deltamethrin residue concentrations (>6 ng/g) were reported in different food items (tomato soup, shredded wheat cereal, whole wheat bread, graham crackers, sugar cookies, and pretzels) [9]. Currently, it remains unclear whether pesticide residue levels differ substantially in commonly consumed food items of US adults measured in the TDS compared to ‘real world’ studies and more research is warranted.

At the moment, up-to-date oral reference doses (RfDs) are not available for these target insecticides in the US EPA’s Integrated Risk Management System (IRIS) or in the US EPA’s Office of Pesticide Program (OPP) database system [27]. In the absence of publicly available risk values for humans dietarily exposed to pyrethroids, Wolansky et al. [28] was used to derive oral RfD values (ng/kg/day) for each target pyrethroid using the following equation: RfD = TD/UF. In this equation, TD (threshold dose [ng/kg/day] was the highest oral dose by gavage that had no observed effect on the motor activity of adult, male Long Evans rats) was divided by UF (uncertainty factor of 100 [unitless] to account for both intra-species and intra-species differences). The calculated oral RfD values were 12,800 ng/kg/day (bifenthrin), 8800 ng/kg/day (cyfluthrin), 5200 ng/kg/day (cyhalothrin), 42,600 ng/kg/day (cypermethrin), 9900 ng/kg/day (deltamethrin), 4800 ng/kg/day (esfenvalerate), 169,900 ng/kg/day (total permethrin). In comparison to previous work [13], using all the DDSF samples (*with available food mass records [g]*), the Ex-R adults’ estimated maximum dietary intake doses over a 24 h period were 63.1 ng/kg/day (bifenthrin), 265 ng/kg/day (cyfluthrin), 69.7 ng/kg/day (cyhalothrin), 393 ng/kg/day (cypermethrin), 2.93 ng/kg/day (deltamethrin), 1784 ng/kg/day (esfenvalerate), and 2115 ng/kg/day (total permethrin). Based on these data, the adults’ estimated maximum dietary intake doses were between 2.7 (esfenvalerate) and 3379 times (deltamethrin) lower than the derived oral RfDs. As such, the estimated maximum dietary intake doses for these pyrethroids are considered below a level of health concern. However, based on these DDSF samples, it was unclear which consumed, single solid food items likely contributed the greatest to the adults’ estimated dietary intake doses to these insecticides. In this current work, the single solid food samples that contributed the highest (top 10%) to the adults’ dietary intake doses (>30 ng/kg/day) to these pyrethroids were:Bifenthrin: Fruit smoothie with milk, vegetable burrito bowl, stir fry with quinoa, and plain cereal with milkCyfluthrin: Chicken pizzaCyhalothrin: Tofu with vegetables and plain cereal with milkCypermethrin: Fruit smoothie with milk, egg and sausage biscuit, and chicken biscuitDeltamethrin: Chicken leg and cakeEsfenvalerate: Yogurt with cereal and chicken buffalo sandwichPermethrin: Pizza, lasagna, and stir fry with noodles

In conclusion, the Ex-R adults were dietarily exposed to single and multiple pyrethroid insecticides in a variety of solid food items they consumed in their everyday environments. An advantage of this type of study design is that it accounts for possible increases or decreases in pyrethroid residue levels in food items that are prepared and/or eaten at home or other settings (i.e., restaurants or work). This research has provided exposure modelers and risk assessors with real-world data, which were lacking in the literature, on the specific solid food items that may be substantially contributing to the aggregate and cumulative dietary exposures of US adults to pyrethroid insecticides.

## Figures and Tables

**Figure 1 toxics-08-00006-f001:**
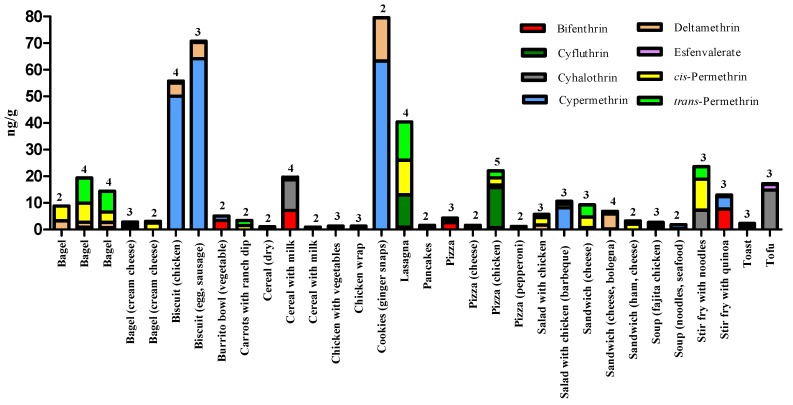
The co-occurrence of the target pyrethroids in 33 different duplicate-single solid food items ^ab^. ^a^ The greatest maximum value (170.1 ng/g) was found in a Greek pizza sample (excluded from this table because the value was at least twice as high as the next highest value); this item contained only *cis*-permethrin (96.4 ng/g) and *trans*-permethrin (73.7 ng/g). ^b^ The number of pyrethroids detected in each food item is shown above each bar in the graph.

**Table 1 toxics-08-00006-t001:** Pyrethroid residue levels (ng/g) that were found in 73 different solid food items consumed by Ex-R adults.

Food Item ^a^	Contained a Fruit or Vegetable	Bifenthrin	Cyfluthrin	Cyhalothrin	Cypermethrin	Deltamethrin	Esfenvalerate	*cis*-Permethrin	*trans*-Permethrin
Bagel	Raisins	---	---	---	---	3.3	---	5.5	---^b^
Bagel	Raisins and tomatoes	0.88	---	---	---	1.9	---	7.1	9.5
Bagel	Raisins and tomatoes	0.68	---	---	---	2.0	---	3.9	7.8
Bagel (cream cheese)	Tomatoes	---	---	---	---	0.85	---	0.99	0.98
Bagel (cream cheese)		---	---	---	---	---	---	2.4	0.68
Bagel (egg)	Blueberries	---	---	---	3.5	---	---	---	---
Biscuit		---	---	---	---	12.0	---	---	---
Biscuit (chicken)		---	---	---	50.1	4.9	---	0.29	0.45
Biscuit (egg and sausage)		---	---	---	64.2	6.1	---	0.42	---
Biscuit (sausage)		0.32	---	---	---	---	---	---	---
Burrito bowl (vegetable)	Peppers, onions, corn, avocados, tomatoes, and beans	3.4	---	---	1.7	---	---	---	---
Cake		---	---	---	---	8.9	---	---	---
Cake (chocolate)		6.8	---	---	---	---	---	---	---
Carrots with ranch dip	Carrots	---	---	---	---	---	---	1.5	1.9
Cereal (dry)		---	---	---	---	---	---	0.38	0.64
Cereal with milk		7.2	---	11.5	---	---	---	0.75	---
Cereal with milk		---	---	---	---	---	---	0.57	---
Cereal with milk		---	---	---	---	---	---	0.33	0.56
Cereal with milk	Cranberries	---	---	---	---	0.96	---	---	---
Cereal with milk	Cranberries	---	---	---	---	2.6	---	---	---
Cereal with milk (soy)		0.29	---	---	---	---	---	---	---
Cereal bar	Strawberries	---	---	---	---	1.6	---	---	---
Chicken with vegetables	Tomatoes, yellow zucchini, broccoli, and carrots	0.51	---	---	---	---	---	0.33	0.45
Chicken leg		---	---	---	---	6.6	---	---	---
Chicken wrap	Lettuce, onions, and green peppers	0.68	---	---	---	---	---	0.33	0.28
Cookie	Apricots	---	---	---	---	---	---	---	0.38
Cookies (caramel)	Coconuts	---	---	---	---	3.1	---	---	---
Cookies (ginger snaps)		---	---	---	63.3	16.3	---	---	---
Donuts (chocolate powdered sugar)		---	---	---	---	3.0	---	---	---
Fruit (mixed)	Pineapples and strawberries	---	---	---	---	---	---	1.3	---
Fruit smoothie with milk (coconut)	Banana and blueberries	---	---	---	29.3	---	---	---	---
Fruit smoothie with milk (coconut)	Banana, pineapple, strawberries, and mangoes	13.9	---	---	---	---	---	---	---
Lasagna	Tomatoes	0.53	---	---	---	---	---	---	---
Lasagna	Spinach and tomatoes	0.93	12.1	---	---	---	---	13.0	14.4
Noodles		---	---	---	1.7	---	---	---	---
Oatmeal		---	---	---	---	---	---	0.32	---
Oatmeal with milk (rice)	Cranberries	---	---	---	---	---	---	---	0.27
Pancakes	Strawberries	---	---	---	---	---	---	0.77	0.80
Pastry		---	---	---	---	2.8	---	---	---
Pizza	Tomatoes, peppers, and onions	2.6	---	---	---	---	---	0.72	1.0
Pizza (cheese)	Tomatoes	0.38	---	---	---	1.2	---	---	---
Pizza (cheese)	Tomatoes	0.69	---	---	---	---	---	---	---
Pizza (chicken)	Spinach and tomatoes	0.75	15.0	1.0	---	---	---	2.7	2.6
Pizza (Greek)	Spinach, tomatoes, olives, and onions	---	---	---	---	---	---	96.4	73.7
Pizza (pepperoni)	Tomatoes	0.69	---	---	---	---	---	0.45	---
Potato chips (barbeque flavor)	Potatoes	---	---	---	---	---	---	---	0.44
Pretzels		---	---	---	---	1.1	---	---	---
Salad with chicken	Lettuce, cucumbers, and tomatoes	---	---	---	---	1.7	---	2.8	1.2
Salad with chicken	Onions, greens, carrots, cucumbers, and orange/red peppers	1.9	---	---	---	---	---	---	---
Salad with chicken (barbeque)	Lettuce, celery, and carrots	---	---	---	8.2	---	---	1.3	1.1
Sandwich (egg and bacon)		---	---	---	---	0.66	---	---	---
Sandwich (egg, bacon, and cheese)	Green peppers and onions	---	---	---	---	2.5	---	---	---
Sandwich (egg and ham)		---	---	---	---	0.89	---	---	---
Sandwich (egg omelet)	Green/red peppers and onions	---	---	---	---	0.77	---	---	---
Sandwich (cheese)	Tomatoes	---	---	---	---	0.68	---	4.0	4.6
Sandwich (cheese and bologna)		0.22	---	---	---	5.6	---	0.30	0.71
Sandwich (chicken buffalo)	Lettuce, tomatoes, pickles, banana peppers, black olives, and jalapenos	---	---	---	---	---	4.0	---	---
Sandwich (ham and cheese)	Lettuce and pickles	---	---	---	---	---	---	2.0	1.2
Sandwich (peanut butter and jelly)	Strawberries	0.93	---	---	---	---	---	---	---
Sandwich (peanut butter and jelly)	Strawberries	0.99	---	---	---	---	---	---	---
Soup (fajita chicken)	Onions, kidney beans, red peppers, jalapenos, tomatoes, carrots, corn, and celery	0.37	---	---	---	---	---	1.2	1.1
Soup (noodles and seafood)	Bean sprouts and jalapenos	0.45	---	---	1.4	---	---	---	---
Spaghetti	Tomatoes	0.60	---	---	---	---	---	---	---
Spaghetti	Tomatoes	0.88	---	---	---	---	---	---	---
Stir fry with noodles	Broccoli, yellow carrots, red peppers, and soybeans	---	---	7.3	---	---	---	11.6	4.7
Stir fry with quinoa	Kale and cranberries	7.7	---	---	4.7	---	---	---	0.54
Toast		---	---	---	---	0.81	---	0.71	0.79
Tofu with vegetables	Kale, onions, cranberries, and soybeans	---	---	14.8	---	---	2.1	0.33	---
Yogurt	Blackberries	7.3	---	---	---	---	---	---	---
Yogurt	Strawberries	---	---	---	---	---	1.7	---	---
Yogurt with cereal		---	---	---	---	---	---	0.33	---
Yogurt with cereal		0.22	---	---	---	---	---	---	---
Yogurt with cereal (bran)	Black cherries and blueberries	---	---	---	---	---	16.7	---	---

^a^ Note, US terminology was used to name individual food items (e.g., biscuit); ^b^ Isomer concentration not reported because this sample did not pass laboratory quality control standards.

**Table 2 toxics-08-00006-t002:** Comparison of pyrethroid residue levels (ng/g) in food items with and without any fruits and/or vegetables.

Pyrethroid ^a,b,c^	Food Item (With a Fruit/Vegetable)		Food Item (Without a Fruit/Vegetable)		*p*-Value ^e^
	N ^d^	Median	N	Median	
Bifenthrin	22	0.82	6	0.31	0.123
Deltamethrin	13	1.7	13	4.9	0.058
*cis*-Permethrin	21	1.5	11	0.38	0.001
*trans*-Permethrin	21	1.1	6	0.66	0.102

^a^ Pyrethroids detected ≥14% in the food samples were used in this analysis. ^b^ Only values ≥LOQ for each pyrethroid were used in this analysis. ^c^ An unpaired Mann–Whitney test was used to compare median levels of a pyrethroid between food groups. ^d^ Number of samples. ^e^ Statistically significant (*p* < 0.05).

**Table 3 toxics-08-00006-t003:** The adult’s estimated dietary exposures and intake doses for the individual pyrethroids (≥LOQ) for each food item.

Food Item	Concentration (ng/g)	Food Mass ^a^ (g)	Exposure (ng/day)	Intake Dose (ng/kg/day)
*cis-*Permethrin (*n* = 32)				
Pizza (Greek)	96.4	488	47,040	532
Lasagna	13.0	170	2200	22.9
Stir fry with noodles	11.6	238	2770	31.5
Bagel	7.1	82.9	589	10.3
Bagel	5.5	38.2	211	3.7
Sandwich (cheese)	4.0	132	528	9.2
Bagel	3.9	98	383	6.7
Salad with chicken	2.8	307	866	11.5
Pizza (chicken)	2.7	323 ^b^	863	15.0
Bagel (cream cheese)	2.4	57.3 ^b^	136	2.4
Sandwich (ham, cheese)	2.0	180	357	3.2
Carrots (ranch dip)	1.5	185	277	3.3
Salad with chicken (barbeque)	1.3	187	241	3.0
Fruit (mixed)	1.3	215	277	3.3
Soup (fajita chicken)	1.2	519	626	7.1
Bagel (cream cheese)	0.99	50.9	50.6	0.88
Pancakes	0.77	255	196	2.0
Cereal with milk	0.75	299	225	3.2
Pizza	0.72	481	347	4.3
Toast	0.71	18.8	13.3	0.23
Cereal with milk	0.57	280	158	2.2
Pizza (pepperoni)	0.45	149	66.3	0.68
Biscuit (egg and sausage)	0.42	129	53.8	0.48
Cereal (dry)	0.38	58.6	22.2	0.31
Tofu	0.33	644	213	2.2
Chicken and vegetables	0.33	248	82.2	0.94
Yogurt with cereal	0.33	189	61.7	0.47
Cereal with milk	0.33	167	54.5	0.70
Chicken wrap	0.33	161	52.7	0.52
Oatmeal	0.32	149	48.0	1.0
Sandwich (cheese and bologna)	0.30	62.7 ^b^	18.5	0.19
Biscuit (chicken)	0.29	156	44.7	0.40
*trans-*Permethrin (*n* = 27)				
Pizza (Greek)	73.7	488	36,980	407
Lasagna	14.4	170	2440	25.4
Bagel	9.5	82.9	787	13.8
Bagel	7.8	97.6	757	13.2
Stir fry with noodles	4.7	238	1110	12.7
Sandwich (cheese)	4.6	132	602	10.5
Pizza (chicken)	2.6	323 ^b^	850	14.8
Carrots (ranch dip)	1.9	185	353	4.2
Sandwich (ham and cheese)	1.2	156	195	1.7
Salad with chicken	1.2	307	371	4.9
Salad with chicken (barbeque)	1.1	187	210	2.6
Soup (fajita chicken)	1.1	519	596	6.8
Pizza	1.0	481	483	6.0
Bagel (cream cheese)	0.98	50.9	49.6	0.87
Pancakes	0.80	255	204	2.1
Toast	0.79	18.8	14.8	0.26
Sandwich (cheese and bologna)	0.71	62.7 ^b^	44.4	0.45
Bagel (cream cheese)	0.68	57.3 ^b^	38.7	0.67
Cereal (dry)	0.64	58.6	37.7	0.53
Cereal with milk	0.56	167	94.0	1.2
Stir fry with quinoa	0.54	396	212	2.2
Biscuit (chicken)	0.45	156	70.6	0.63
Chicken with vegetables	0.45	248	111	1.3
Potato chips (barbeque flavor)	0.44	27.4	12.1	0.11
Cookie	0.38	17.3	6.54	0.14
Chicken wrap	0.28	161	44.9	0.44
Oatmeal with milk (rice)	0.27	183	50.0	0.81
Bifenthrin (*n* = 28)				
Fruit smoothie with milk (coconut)	13.9	421	5840	59.4
Stir fry with quinoa	7.7	396	3050	31.0
Yogurt	7.3	89.4 ^b^	653	10.7
Cereal with milk	7.2	299	2160	30.3
Cake (chocolate)	6.8	83.8	570	6.5
Burrito bowl (vegetable)	3.4	580	1990	34.3
Pizza	2.6	481	1260	15.6
Salad with chicken	1.9	258	483	5.5
Sandwich (peanut butter and jelly)	0.99	107	106	1.2
Lasagna	0.93	170	158	1.7
Sandwich (peanut butter and jelly)	0.93	151 ^b^	141	1.8
Bagel	0.88	82.9	73.3	1.3
Spaghetti	0.88	484	427	5.3
Pizza (chicken)	0.75	323 ^b^	243	4.2
Pizza (cheese)	0.69	126	86.8	0.91
Pizza (pepperoni)	0.69	149	103	1.0
Bagel	0.68	97.6	65.9	1.1
Chicken wrap	0.68	161	109	1.1
Spaghetti	0.60	80.6 ^b^	48.5	0.64
Lasagna	0.53	198	105	1.3
Chicken and vegetables	0.51	248	127	1.5
Soup (noodles and seafood)	0.45	524	236	3.8
Pizza (cheese)	0.38	248	93.9	0.94
Soup (fajita chicken)	0.37	519	192	2.2
Biscuit (sausage)	0.32	110	35.7	0.27
Cereal with milk (soy)	0.29	254	73.6	0.91
Sandwich (cheese and bologna)	0.22	62.7 ^b^	13.5	0.14
Yogurt with cereal	0.22	186	40.2	0.31
Deltamethrin *(n* = 26)				
Cookies (ginger snaps)	16.3	10.6	173	2.5
Biscuit	12.0	88.2	1060	16.4
Cake	8.9	374 ^b^	3330	33.5
Chicken leg	6.6	353 ^b^	2320	38.1
Biscuit (egg and sausage)	6.1	129	777	6.9
Sandwich (cheese and bologna)	5.6	62.7 ^b^	350	3.5
Biscuit (chicken)	4.9	156	770	6.9
Bagel	3.3	38.2	128	2.2
Cookies (caramel)	3.1	40.5	125	0.96
Doughnuts (chocolate powdered sugar)	3.0	166	505	6.7
Pastry	2.8	39.9	111	1.7
Cereal with milk	2.6	245	627	7.1
Sandwich (egg, bacon, and cheese)	2.5	98.0	249	3.3
Bagel	2.0	97.6	195	3.4
Bagel	1.9	82.9	160	2.8
Salad with chicken	1.7	307	507	6.8
Cereal bar	1.6	153 ^b^	247	2.8
Pizza (cheese)	1.2	248	300	3.0
Pretzels	1.1	12.8	13.5	0.14
Cereal with milk	0.96	112 ^b^	108	2.0
Sandwich (egg and ham)	0.89	135	120	1.1
Bagel (cream cheese)	0.85	50.9	43.3	0.76
Toast	0.81	18.8	15.3	0.27
Sandwich (egg omelet)	0.77	75.7	58.0	0.77
Sandwich (cheese)	0.68	132	90.2	1.6
Sandwich (egg and bacon)	0.66	127	84.0	0.75
Cyfluthrin (*n* = 2)				
Pizza (chicken)	15.0	323 ^b^	4860	84.4
Lasagna	12.1	170	2050	21.4
Cyhalothrin (*n* = 4)				
Tofu with vegetables	14.8	644	9560	97.1
Cereal with milk	11.5	299	3440	48.4
Stir fry with noodles	7.3	238	1730	19.8
Pizza (chicken)	1.0	323 ^b^	324	5.6
Cypermethrin (*n* = 10)				
Biscuit (egg and sausage)	64.2	129	8250	73.7
Cookies (ginger snaps)	63.3	10.6	671	9.6
Biscuit (chicken)	50.1	156	7810	69.7
Fruit smoothie with milk (coconut)	29.3	329	9660	98.2
Salad with chicken (barbeque)	8.2	187	1530	18.7
Stir fry with quinoa	4.7	396	1850	18.8
Bagel (egg)	3.5	81.9	288	3.8
Burrito bowl (vegetable)	1.7	580	978	16.8
Noodles	1.7	491	845	9.1
Soup (noodles and seafood)	1.4	524	723	11.7
Esfenvalerate (*n* = 4)				
Yogurt with cereal (bran)	16.7	175	2920	40.1
Sandwich (chicken buffalo)	4.0	521 ^b^	2060	35.7
Tofu	2.1	644	1380	14.0
Yogurt	1.7	195	331	3.4

^a^ The amount of a food item was determined by measuring the weight of food (g) in the individual sampling bags; ^b^ The amount of a food item was determined by using the recorded amount (cups) in the participants’ food diaries.

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
