# Peer review of "Dietary Pyrethroid Exposures and Intake Doses for 188 Duplicate-Single Solid Food Items Consumed by North Carolina Adults"

_toxics, 2020, doi:10.3390/toxics8010006_

Round 1

Reviewer 1 Report

General comments

The study provides valuable information on real-world contamination of various food items with pesticide residues. The analytical method is valid and the results are not doubted. However, the manuscript should be updated by additional information of which a part (in my view) must be given whereas the article, in general, could  benefit very much from further data and considerations even though it might be not essential to include them.

The following information appears mandatory to amend:

The group of 50 participants should be described in greater detail (including body weigth distribution, if available). One is missing an opinion if these people may be considered representative for, e.g., the population in NC or for that part of the NC population living in a city.

For the LODs, it should be reported how they were determined. Even more important, it is absolutely necessary to give the "limit of quantification" (LOQ) for each pesticide. This is the much more relevant parameter than the LOD. Only samples containing residues equal to or above the LOQ may be used for any calculation.

The article could benefit very much from:

(1) a comparison of the estimated daily intakes with acute reference doses as set by EPA for the different compounds. If the same person was exposed to a variety of residues, the percentages at which the different aRfDs were covered could be summed up (i.e., intake of bifenthrin accounting for 22% of aRfD + intake of deltamethrin accounting for 37% of aRfD + ...) to reveal if there is a "cumulative risk".

(2) a comparison with data obtained in TDS, if possible.

(3) a comparison of estimated daily intakes with model calculations which undoubtely exist and should be on file at EPA. This could help to to check if the model assumptions were sufficiently conservative.

Specific comments

Introduction, lines 38-44

It may be useful to explain a bit better the merits of this study as compared to the TDS. At first, it was my understanding that an impact of processing of food before consumption was included here but when food items purchased from stores or supermarkets were also "prepared as 'consumed'", there is no much difference left between the study conditions since food was always processed. Why should we consider the actual study more realistic or more robust then the rather large TDS?

Author Response

Reviewer 1

General comments

The study provides valuable information on real-world contamination of various food items with pesticide residues. The analytical method is valid and the results are not doubted. However, the manuscript should be updated by additional information of which a part (in my view) must be given whereas the article, in general, could benefit very much from further data and considerations even though it might be not essential to include them.

The following information appears mandatory to amend:

The group of 50 participants should be described in greater detail (including body weigth distribution, if available). One is missing an opinion if these people may be considered representative for, e.g., the population in NC or for that part of the NC population living in a city.

Response: I agree with the reviewer. In the Materials and Methods section under subsection 2.1. Study Background, the following sentences were added to provide more information about the study cohort - “Adult participants were recruited by an EPA contractor using an existing database of potential volunteers or by word of mouth (e.g., previous study participants) [13]. A total of 30 females (ages 21–50) and 20 males (ages 19–48) were recruited [13]. The data presented in this article apply only to this study cohort and cannot be generalized to all adults in North Carolina or all adults in the US.”

Also, the following sentence was added to subsection 2.5 Estimation of Dietary Exposures and Intake Doses per Solid Food Item - Body weights of the females ranged from 48.1–130 kg and, body weights of the males ranged from 57.6–130 kg [13].

For the LODs, it should be reported how they were determined. Even more important, it is absolutely necessary to give the "limit of quantification" (LOQ) for each pesticide. This is the much more relevant parameter than the LOD. Only samples containing residues equal to or above the LOQ may be used for any calculation.

Response: I agree with the reviewer. I have revised this paragraph as follows: “The limits of quantitation (LOQs) were 0.10 ng/g (bifenthrin), 2.5 ng/g (cyfluthrin), 1.0 ng/g (cyhalothrin), 1.0 ng/g (cypermethrin), 0.25 ng/g (deltamethrin), 1.0 ng/g (esfenvalerate), 0.25 ng/g (cis-permethrin), and 0.25 ng/g (trans-permethrin), respectively. The LOQs were based on the levels of the lowest calibration standards with a signal-to-noise ratio of not less than 3 [13]. For these analytes, the estimated limits of detection were 2 to 5 times lower than the estimated LOQs [13]. “In addition, only values > LOQ per analyte were used for the statistical analyses (see the Statistical Analysis subsection); I have updated the entire manuscript, as needed, based on these values.

The article could benefit very much from:

(1) a comparison of the estimated daily intakes with acute reference doses as set by EPA for the different compounds. If the same person was exposed to a variety of residues, the percentages at which the different aRfDs were covered could be summed up (i.e., intake of bifenthrin accounting for 22% of aRfD + intake of deltamethrin accounting for 37% of aRfD + ...) to reveal if there is a "cumulative risk".

Response: I agree with the reviewer and have revised the following paragraph in the Discussion section - “At the moment, up-to-date oral reference doses (RfDs) are not available for these target insecticides in the US EPA’s Integrated Risk Management System (IRIS) or in the US EPA’s Office of Pesticide Program (OPP) database system [27]. In the absence of publicly-available risk values for humans dietarily exposed to pyrethroids, Wolansky et al. [28] was used to derive oral RfD values (ng/kg/day) for each target pyrethroid using the following equation: RfD = TD/UF. In this equation, TD (threshold dose [ng/kg/day] was the highest oral dose by gavage that had no observed effect on the motor activity of adult, male Long Evans rats) was divided by UF (uncertainty factor of 100 [unitless] to account for both intra-species and intra-species differences). The calculated oral RfD values were 12,800 ng/kg/day (bifenthrin), 8,800 ng/kg/day (cyfluthrin), 5,200 ng/kg/day (cyhalothrin), 42,600 ng/kg/day (cypermethrin), 9,900 ng/kg/day (deltamethrin), 4,800 ng/kg/day (esfenvalerate), 169,900 ng/kg/day (total permethrin). In comparison to previous work [13], using all the DDSF samples (with available food mass records [g]), the Ex-R adults’ estimated maximum dietary intake doses over a 24-h period were 63.1 ng/kg/day (bifenthrin), 265 ng/kg/day (cyfluthrin), 69.7 ng/kg/day (cyhalothrin), 393 ng/kg/day (cypermethrin), 2.93 ng/kg/day (deltamethrin), 1784 ng/kg/day (esfenvalerate), and 2115 ng/kg/day (total permethrin). Based on these data, the adults’ estimated maximum dietary intake doses were between 2.7 (esfenvalerate) and 3379 times (deltamethrin) lower than the derived oral RfDs. As such, the estimated maximum dietary intake doses for these pyrethroids are considered below a level of health concern.

(2) a comparison with data obtained in TDS, if possible.

Response: I have added the following paragraph to the Discussion section – “In the US, the FDA’s TDS measures pesticide residues in food items commonly eaten in the average diet of the general population [9]. These data are used to estimate the annual dietary intake of adults to specific pesticides, including pyrethroid insecticides [9]. The TDS purchases about 280 different kinds of food items from grocery stores and supermarkets from four geographical regions of the country each year; the foods are then prepared as consumed and analyzed for pesticides in a laboratory [9]. Few US studies, however, have measured pesticide residues in similar food items consumed by adults in their everyday environments [10,12,13]. This is important as research has indicated that people’s personal behaviors and hygiene (e.g., not washing hands before eating, not cleaning kitchen countertops before preparing foods on them, or not rinsing fresh produce with tap water before consumption) can greatly influence pesticide residue levels found in some foods [12,23,24,25]. For instance, studies have shown that pyrethroid residues have been found on kitchen countertops at homes [24,25], and these residues can be transferable to foods contacting these surfaces [26]. The major benefit of these ‘real world’ studies is that they account for possible increases or decreases in pesticide residue concentrations occurring in food items that are prepared and/or eaten by people at home or other places (i.e., work or school). In this current study, food items with the greatest deltamethrin residues (> 6 ng/g) were gingersnap cookies, cake, a plain biscuit, chicken leg, and an egg and sausage biscuit. However, in comparison to the 2009–2012 TDS surveys, the highest deltamethrin residue concentrations (> 6 ng/g) were reported in different food items (tomato soup, shredded wheat cereal, whole wheat bread, graham crackers, sugar cookies, and pretzels) [9]. Currently, it remains unclear whether pesticide residue levels differ substantially in commonly consumed food items of US adults measured in the TDS compared to ‘real world’ studies and more research is warranted.”

 (3) a comparison of estimated daily intakes with model calculations which undoubtely exist and should be on file at EPA. This could help to to check if the model assumptions were sufficiently conservative.

Response: No change. This is outside the scope of this work. I am also unaware of a file that exists at EPA on the “comparison of estimated daily intakes with model calculations” for the target pyrethroids in single food items.

 Specific comments

Introduction, lines 38-44

It may be useful to explain a bit better the merits of this study as compared to the TDS. At first, it was my understanding that an impact of processing of food before consumption was included here but when food items purchased from stores or supermarkets were also "prepared as 'consumed'", there is no much difference left between the study conditions since food was always processed. Why should we consider the actual study more realistic or more robust then the rather large TDS?

Response: I agree with the reviewer; for clarification, I have reworded the following sentence in the second paragraph of the Introduction section – “This type of study design is important as people’s personal behaviors and hygiene (e.g., not cleaning kitchen countertops before preparing foods on them or not rinsing fresh produce with tap water before consumption) can greatly impact pesticide residue levels in some food items at home or other settings (i.e., restaurants) [12]. At the moment, it is unclear whether pesticide residue levels differ substantially in commonly consumed food items of adults measured in the FDA’s TDS compared to ‘real world’ studies.”  

Reviewer 2 Report

The manuscript describes a reanalysis of dietary exposure data to provide levels of some commonly used pyrethroid insecticide levels present in individual food items. The data is clearly presented and would contribute towards a wider understanding of dietary exposure to pesticides.

I have some minor comments:

Line 162: The author might wish to consider that different regions of the world have different terminology for the same food stuff. The definition of biscuit is certainly different between US and Europe. Consider whether some clarification might be helpful for a worldwide audience.

Line 205: The author uses median level calculated from the samples with detectable amounts of pyrethroid present. I think that the median level for all samples (n=188) would be <LoD. The manuscript clearly states (multiple times) that this methodology is being used, but, perhaps including an additional statistical metric (something like 90th percentile of the total data set) would give additional context to the dataset?

Discussion:  It might be helpful to add a short discussion of the levels reported in this manuscript in relation to acceptable daily intakes.

Author Response

Reviewer 2

The manuscript describes a reanalysis of dietary exposure data to provide levels of some commonly used pyrethroid insecticide levels present in individual food items. The data is clearly presented and would contribute towards a wider understanding of dietary exposure to pesticides.

I have some minor comments:

Line 162: The author might wish to consider that different regions of the world have different terminology for the same food stuff. The definition of biscuit is certainly different between US and Europe. Consider whether some clarification might be helpful for a worldwide audience.

Response: This is a good point, but it would not be practical to list all the names for the same food items (e.g., biscuit vs. cookie) used in other countries around the world. For clarification, I have added the following footnote to Table 1: aNote US terminology was used to name individual food items (e.g., biscuit)."

Line 205: The author uses median level calculated from the samples with detectable amounts of pyrethroid present. I think that the median level for all samples (n=188) would be <LoD. The manuscript clearly states (multiple times) that this methodology is being used, but, perhaps including an additional statistical metric (something like 90th percentile of the total data set) would give additional context to the dataset?

Response: Based on a comment made by a previous reviewer, I have calculated all statistics in the manuscript now based on the LOQ, not the LOD. I have added the following sentence in the Results section under subsection 3.1. Pyrethroid Residue Levels in the Single Food Items as Consumed: “Using all 188 samples, residue levels for these four pyrethroids at the 90th percentile were 0.72 (cis-permethrin), 0.64 ng/g (trans-permethrin), 0.61 ng/g (bifenthrin), and 1.1 ng/g (deltamethrin).” – these values were derived based on values > LOQ per analyte.

Discussion:  It might be helpful to add a short discussion of the levels reported in this manuscript in relation to acceptable daily intakes.

Response: I agree with the reviewer and have revised the following paragraph in the Discussion section - “At the moment, up-to-date oral reference doses (RfDs) are not available for these target insecticides in the US EPA’s Integrated Risk Management System (IRIS) or in the US EPA’s Office of Pesticide Program (OPP) database system [27]. In the absence of publicly-available risk values for humans dietarily exposed to pyrethroids, Wolansky et al. [28] was used to derive oral RfD values (ng/kg/day) for each target pyrethroid using the following equation: RfD = TD/UF. In this equation, TD (threshold dose [ng/kg/day] was the highest oral dose by gavage that had no observed effect on the motor activity of adult, male Long Evans rats) was divided by UF (uncertainty factor of 100 [unitless] to account for both intra-species and intra-species differences). The calculated oral RfD values were 12,800 ng/kg/day (bifenthrin), 8,800 ng/kg/day (cyfluthrin), 5,200 ng/kg/day (cyhalothrin), 42,600 ng/kg/day (cypermethrin), 9,900 ng/kg/day (deltamethrin), 4,800 ng/kg/day (esfenvalerate), 169,900 ng/kg/day (total permethrin). In comparison to previous work [13], using all the DDSF samples (with available food mass records [g]), the Ex-R adults’ estimated maximum dietary intake doses over a 24-h period were 63.1 ng/kg/day (bifenthrin), 265 ng/kg/day (cyfluthrin), 69.7 ng/kg/day (cyhalothrin), 393 ng/kg/day (cypermethrin), 2.93 ng/kg/day (deltamethrin), 1784 ng/kg/day (esfenvalerate), and 2115 ng/kg/day (total permethrin). Based on these data, the adults’ estimated maximum dietary intake doses were between 2.7 (esfenvalerate) and 3379 times (deltamethrin) lower than the derived oral RfDs. As such, the estimated maximum dietary intake doses for these pyrethroids are considered below a level of health concern.

Round 2

Reviewer 1 Report

The author did a lot to improve the quality of this publication. I feel that the significance of this research is more clear now to the reader. As presented now, the data might be of interest also for regulatory purposes. I have no additional comments, questions or recommendations and I suggest publication of the manuscript in its revised version.